# Cut out and Replay: A Simple yet Versatile Strategy for Multi-Label Online Continual Learning

**Xinrui Wang** [1 2]  **Shao-Yuan Li** [1 2 3]  **Jiaqiang Zhang** [1 2]  **Songcan Chen** [1 2]

## Abstract

Multi-Label Online Continual Learning (MOCL) requires models to learn continuously from endless multi-label data streams, facing complex challenges including persistent catastrophic forgetting, potential missing labels, and uncontrollable imbalanced class distributions. While existing MOCL methods attempt to address these challenges through various techniques, *they all overlook label-specific region identifying and feature learning* - a fundamental solution rooted in multi-label learning but challenging to achieve in the online setting with incremental and partial supervision. To this end, we first leverage the inherent structural information of input data to evaluate and verify the innate localization capability of different pre-trained models. Then, we propose CUTER (CUT-out-and-Experience-Replay), a simple yet versatile strategy that provides fine-grained supervision signals by further identifying, strengthening and cutting out label-specific regions for efficient experience replay. It not only enables models to simultaneously address catastrophic forgetting, missing labels, and class imbalance challenges, but also serves as an orthogonal solution that seamlessly integrates with existing approaches. Extensive experiments on multiple multi-label image benchmarks demonstrate the superiority of our proposed method. The code is available at https://github.com/wxr99/Cut-Replay

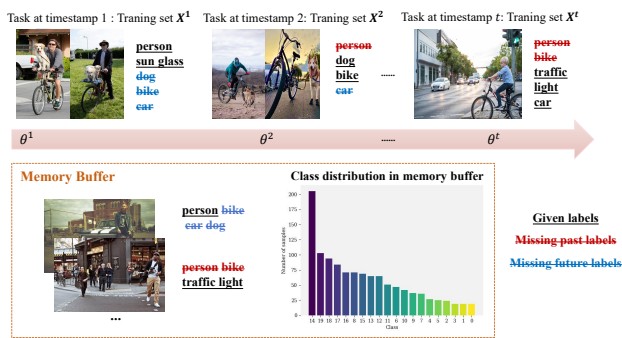

*Figure 1.* Two unique challenges in MOCL compared with traditional OCL: (1) Massive missing past and future labels in both coming data stream and memory buffer. (2) Severe class imbalance that persists in the memory buffer even with re-balancing strategies like CEBS(Wei & Li, 2019a; Yan et al., 2021).

---

[1]College of Computer Science and Technology, Nanjing University of Aeronautics and Astronautics, Nanjing 211106 [2]MIIT Key Laboratory of Pattern Analysis and Machine Intelligence, Nanjing University of Aeronautics and Astronautics, Nanjing 211106 [3]The State Key Laboratory of Novel Software Technology, Nanjing University, P.R. China. Correspondence to: Songcan Chen <s.chen@nuaa.edu.cn>.

*Proceedings of the 42$^{nd}$ International Conference on Machine Learning*, Vancouver, Canada. PMLR 267, 2025. Copyright 2025 by the author(s).

## 1. Introduction

Online continual learning (OCL) enables models to learn from continuous, endless data streams. Significant progress has been made in this area through various techniques to mitigate catastrophic forgetting, such as knowledge distillation, gradient regularization, and experience replay.

However, these OCL approaches focus primarily on single-label classification, while real-world data often exhibit multiple semantic concepts and objects, motivating the study of Multi-Label Online Continual Learning (MOCL) (Kim et al., 2020). Figure 1 shows an illustrative example. At each time step $t$, the training data $D_t$ concerning label space $Y_t$ for task $t$ arrive in a streaming manner. For each example $(x, y) \in D_t$, $y \subset Y_t$ denotes the tagged relevant label set. Given a sequence of $T$ tasks, the objective of MOCL is to efficiently adapt the learning model to the current task while preventing catastrophic forgetting of previously learned tasks.

Compared to single-label scenarios, MOCL faces two specific data challenges: (1) Pervasive missing labels: samples in task $t$ are only annotated with labels from $Y_t$, even when containing objects from old classes $\mathcal{Y}_{1:t-1}$ or future classes

$\mathcal{Y}_{t+1:T}$. These unlabeled classes become *false negatives*, aggravating catastrophic forgetting. (2) Uncontrollable imbalanced classes: the categories often follow a long-tailed distribution. Training on such data would lead the model to be biased towards overfitting the head classes and underfitting the tail classes. The *co-occurrence* of head and tail classes within one sample further complicates this issue.

Several works have attempted to address these challenges. (Kim et al., 2020) first identified the severe forgetting of minority classes in long-tailed scenarios and proposed a Partitioning Reservoir Sampling (PRS) strategy to balance head and tail classes in replay-based approaches. To address the computational inefficiency of PRS, (Liang & Li, 2022) developed Optimizing Class Distribution in Memory (OCDM), which reformulates memory updates as a sample selection optimization problem solvable through a linear-time greedy algorithm. The challenge of missing labels in MOCL was first highlighted by (Du et al., 2022), which introduced an Augmented Graph Convolutional Network (AGCN). This model generates predictions for previously seen classes while modeling dynamic label relationships across sequential tasks and mitigating forgetting through distillation and relationship-preserving losses. Building on this, (Dong et al., 2023) proposed the Knowledge Restore and Transfer (KRT) framework, which combines dynamic pseudo-labeling for old classes with session-specific knowledge transfer. More recently, (Du et al., 2024) tackled both challenges simultaneously through two key components: Asymmetric Knowledge Distillation (AKD) and Online Relabeling (OR). AKD rebalances the learning process by emphasizing negative label learning in classification loss while reducing the impact of overconfident predictions in distillation loss. OR complements this by recovering missing labels in the memory buffer through online relabeling.

Although these works demonstrate promising results, their feature learning mechanisms have inherent limitations with multi-label data. The conventional approach of extracting a single feature vector per example, along with techniques like pseudo-labeling and resampling, suffers from co-occurrence bias between head and tail classes. This limitation is particularly critical with missing labels and class imbalance, where discriminative feature learning requires minimizing interference from label co-occurrence patterns.

With the above understanding, we resort to label-specific feature learning, which was shown to be superior to unified sample-wise features in offline multi-label learning (Zhang & Wu, 2014). Suppose we can successfully identify label-specific regions in images through a straightforward cut-out-and-replay mechanism, i.e., cutting out these regions and storing them in the memory buffer for replay. This mechanism would naturally avoid label co-occurrence interference and missing label issues. Furthermore, with

object-level regions and supervision signals stored in the buffer, it enables more effective experience replay under the same memory overhead, where class imbalance is easily addressed by controlling class distributions in the buffer.

To this end, we propose CUTER (CUT-out-and-Experience-Replay), a simple yet versatile approach that efficiently localizes, strengthens, and cuts out label-specific regions for MOCL with richer fine-grained supervision signals. Motivated by the recent inspiring trials on vision pre-trained models' ability for zero-shot coarse segmentation (Caron et al., 2021; Siméoni et al., 2021; Wang et al., 2023b), we make a thorough study of the pre-trained modes' localization capability. Through extensive empirical validations over several widely used pre-trained models (e.g., DINO, MoCo, MAE), and established theoretical supports from graph theory, we show that the averaged Fiedler value (second smallest eigenvalue of graph Laplacian) of the feature patch similarity graph can serve as a valid evaluation measure for selecting pre-trained models good at localizing precise label-specific regions. The identified salient regions are then selectively stored in a memory buffer based on their prediction confidence and label alignment, effectively transforming multi-label image classification replay into multiple single-label sub-image classification tasks. To further combat the forgetting impact of the model's localization ability during the continual learning process, we draw inspiration from image segmentation principles and incorporate a low-rank constraint on the feature-based similarity adjacency matrix to improve inter-patch separability for more precise region cropping, supported by graph spectral theory.

Serving as an orthogonal solution to exiting OCL models, it enables seamless integration with existing models to address catastrophic forgetting, missing labels, and class imbalance challenges. We conduct extensive experiments on multiple multi-label image benchmarks, showing that our method significantly outperforms state-of-the-art methods in MOCL. In summary, our main contributions are threefold: i) We introduce label-specific learning into MOCL and conduct the first systematic analysis of pre-trained models' potential for MOCL; ii) We propose CUTER, a simple yet versatile replay strategy that simultaneously enhances model performance while addressing catastrophic forgetting, missing labels, and class imbalance; iii) Extensive experiments and ablation studies demonstrate that CUTER not only achieves state-of-the-art performance but also serves as a complementary component that can be seamlessly integrated with existing approaches.

## 2. Method

In this section, we focus on better extracting and utilizing label-specific regions to address MOCL problem. Our method consists of three key steps. First, we propose

an annotation-free evaluation protocol to assess the zero-shot localization potential of popular pre-trained models in MOCL. Second, leveraging the selected pre-trained model and its localization capability, we design a simple yet versatile cut-out-and-replay strategy for effective and efficient region-label matching, cropping, and storage. Finally, to prevent the deterioration of model's localization capability during MOCL progression, we incorporate a regularization term that continuously consolidates and strengthens the model's localization and segmentation abilities.

## 2.1. Zero-Shot Localization Assessment for Pre-trained Models

A key prerequisite for our cut-out-and-replay strategy is identifying suitable pre-trained models through their inherent localization capabilities. Recent studies have shown that advanced vision pre-trained models can naturally localize objects through feature clustering (Caron et al., 2021; Siméoni et al., 2021), making them potential candidates for MOCL. However, conventional evaluation metrics requiring ground truth boxes or masks are impractical, as most multi-label datasets lack such annotations. This motivates our development of an annotation-free evaluation protocol.

To address this challenge, we propose a novel evaluation metric inspired by graph theory and spectral clustering. Our key insight stems from examining popular unsupervised localization methods like Normalized Cut (NCut)(Shi & Malik, 2000), Mask Cut (MCut)(Wang et al., 2023a), and Token Cut (TCut)(Wang et al., 2023b). These methods perform better when feature maps form graphs with stronger separation (weaker connectivity), which is mathematically characterized by the Cheeger constant:

**Definition 2.1.** (Royle & Godsil, 2001) Let $G = (V, E, W)$ be a weighted graph with vertex set $V$, edge set $E$, and edge weight matrix $W$. The Cheeger constant $h(G)$ is defined as:

$$h(G) = \min_{S \subset V} \frac{\mathcal{C}(S, V \setminus S)}{\min\{\mathcal{C}(S, V), \mathcal{C}(V \setminus S, V)\}}$$

where $S$ is a non-empty subset of $V$, $V \setminus S$ represents its complement and $\mathcal{C}(\cdot)$ measures the similarity between two sets, e.g., $\mathcal{C}(A, B) = \sum_{v_i \in A, v_j \in B} W_{ij}$.

Although computing the Cheeger constant is NP-hard, we can leverage its relationship with the graph's Fiedler value as shown in the following Lemma 2.2. Given a sample's feature map $\theta(x)$, we can construct a weighted undirected graph $G = (V, E, A)$ where vertices represent feature vectors of image patches and edge weights $A_{ij} = exp(-\frac{||\theta(x_i) - \theta(x_j)||^2}{2\sigma^2})$ encode patch connectivity using a Gaussian kernel. The Fiedler value $\lambda_2$ is then computed as the second smallest eigenvalue of the graph Laplacian matrix $L = D - A$, where $D$ is an $N \times N$ diagonal degree matrix with elements $d(i) = \sum_j A_{ij}$.

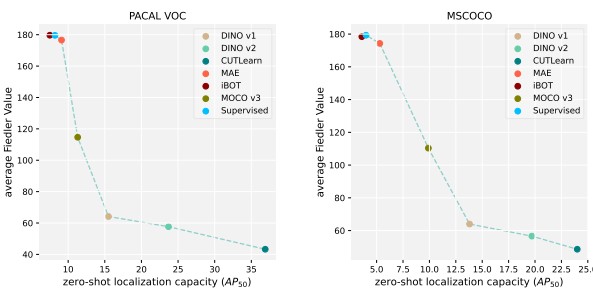

*Figure 2.* Correlation between the averaged Fiedler Value and zero-shot detection performance ($AP_{50}$) on Pascal VOC07 and MSCOCO.

**Lemma 2.2.** *(Chung, 1997) For a weighted undirected graph $G$, let $\lambda_2$ be its Fiedler value, $h(G)$ be its Cheeger constant, and $\Delta = \max_i d(i)$ be the maximum degree in the graph. Then we have:*

$$\frac{\lambda_2}{2} \leq h(G) \leq \sqrt{2\Delta\lambda_2}$$

Based on this theoretical foundation, we propose to assess a model's potential zero-shot localization capability by computing the average Fiedler value of features extracted from a small subset of the downstream dataset. As shown in Figure 2 and Figure 3, they reveal a clear correlation between zero-shot localization performance and the averaged Fiedler value of patch feature graphs. This aligns with graph theory principles: a lower average Fiedler value indicates weaker graph connectivity (Royle & Godsil, 2001), suggesting stronger feature separability and thus better suitability for partition-based operations like MCut and NCut.

Applying this metric to evaluate popular pre-trained models, including MoCo(He et al., 2020; Chen et al., 2020; Chen* et al., 2021), DINO(Caron et al., 2021; Oquab et al., 2023), MAE(He et al., 2022), and iBOT(Zhou et al., 2022), we uncover that multi-crop consistency training significantly enhances innate localization ability. In contrast, reconstruction-based training inherently encourages feature sharing during recovery, potentially limiting the effectiveness of spectral clustering-based localization methods. We leave the detailed analysis in Appendix D. In summary, when partial downstream task data is available, we can evaluate models by computing their average Fiedler value. However, when facing completely unknown future tasks, selecting a model pre-trained with multi-crop consistency is likely a more reliable choice.

## 2.2. Selective Replay via Label-Region Matching

After selecting the initialization model based on the proposed average Fiedler value, the next obstacle for implementing our envisioned cut-and-replay strategy lies in estab-

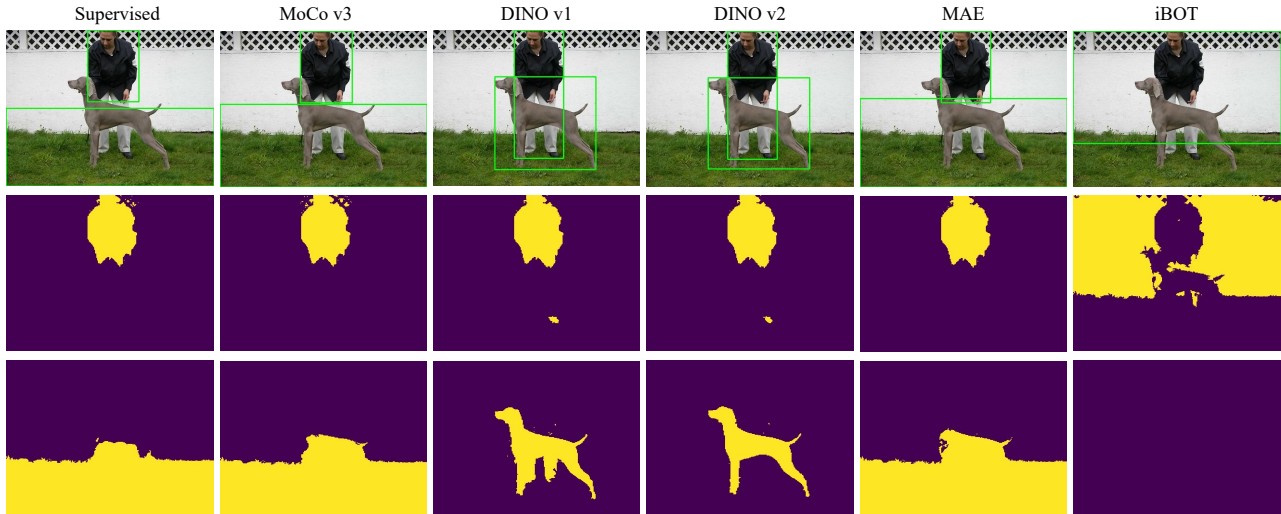

*Figure 3.* Visual comparison of detection (coarse bounding boxes) and segmentation (coarse masks) capabilities across pre-trained models using ViT-S/16 backbone, obtained via two-round MaskCut (Wang et al., 2023a).

lishing precise one-to-one correspondence between image region and its associated label.

$$L_{asl} = \frac{1}{|\mathcal{C}_k|} \sum_{c=1}^{|\mathcal{C}_k|} \begin{cases} (1-p_c)^{\gamma^+} log(p_c), & y_c = 1, \\ p_c^{\gamma^-} log(1-p_c), & y_c = 0, \end{cases} \quad (1)$$

To achieve this goal, for each incoming data pair $(x, y)$, we make the prediction $p = f(x)$ and apply the asymmetric loss (Eq.1) like most MOCL methods, where $y_c$ represents the binary label indicating the presence of class $c$, and $\gamma^+$, $\gamma^-$ denote the positive and negative focusing parameters respectively. Subsequently, we employ MCut to generate a list of binary masks $\{m^j\}_{j=1}^N$ that identify potential foreground objects, where $N$ is a hyper-parameter determining the number of MCut iterations (MCut's detailed background and procedure is provided in Appendix B). These binary masks are then used to derive bounding boxes $\{(h_1^j, w_1^j, h_2^j, w_2^j)\}_{j=1}^N$, enabling us to extract $N$ potential foreground objects $\{x_{obj}^j\}_{j=1}^N$ from the input image $x$.

The extracted objects $\{x_{obj}^j\}$ are then resized and fed into the classification model again to obtain foreground object prediction $p_{obj}^j = f(x_{obj}^j)$. Our experimental results in Table 6 (CUTER vs CUTER w/ Fixed Backbone) demonstrate that for object localization in MOCL, using a model trained with asymmetric loss (Eq.1) (Ridnik et al., 2021) outperforms using a fixed pre-trained backbone like DINO, showing superior performance in identifying current task classes ($\mathcal{C}^k$) and localizing regions corresponding to given labels. After making the cut out through this continuously

updated model, we aim to replay these extracted objects instead of the entire image as done in previous works.

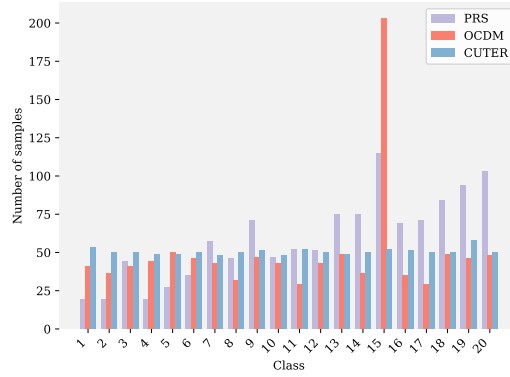

*Figure 4.* Class distribution in the memory buffer (size=1000) for different re-balancing methods after training on VOC dataset.

This storage process of $\{x_{obj}^j\}_{j=1}^N$ consists of two steps. First, to establish reliable label-region correspondences, we retain only the objects that maintain high classification confidence post-resizing and correspond to a single label. Specifically, for a prediction $p_{obj}^j$, we select extracted object $x_{obj}^j$ into the candidate set for the memory buffer if:

$$p_{obj,(1)}^j > \tau \wedge p_{obj,(2)}^j < 0.5 \quad (2)$$

where $p_{obj,(k)}^j$ denotes the $k$-th largest element in $p_{obj}^j$ and $\tau$ is the confidence threshold. Meanwhile, to balance the class distribution in the memory buffer, we employ two thresholds

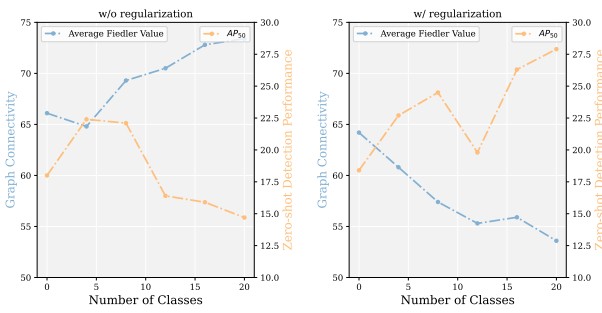

*Figure 5.* Visualization of model's zero-shot localization capability on PASCAL VOC dataset during MOCL training.

$\tau_1$ and $\tau_2$ ($\tau_1 < \tau_2$) based on class frequency. For any class with frequency less than half of the most frequent class, we use the lower threshold $\tau_1$, while assigning $\tau_2$ to the others.

Second, we adopt a modified rebalanced reservoir sampling strategy for the memory buffer. For a new candidate object, its sampling probability is $1 - m/m_{max}$, where $m$ denotes the number of samples with predicted label $y = argmax(p_{obj}^j)$ in the memory buffer, and $m_{max}$ represents the count of the most frequent class. To maintain class balance, we randomly remove samples from the most frequent class when the buffer is full. This straightforward sampling strategy achieves better class balance than OCDM(Liang & Li, 2022) and PRS(Kim et al., 2020) with lower computational overhead, as shown in Figure 4. By storing only regions with single-label correspondence, we enable direct control over class distribution while preserving more discriminative information with reduced memory cost.

### 2.3. Localization-Aware Feature Regularization

Thus far, we have introduced our cut-and-replay strategy for preserving single-label regions and their corresponding features. While this strategy effectively maintains model's classification performance on past classes through a memory-efficient replay mechanism and demonstrates strong adaptability to new classes, we observe that the inherited localization capability from pre-trained models gradually deteriorates during MOCL progression - a phenomenon analogous to the catastrophic forgetting in continual learning. As illustrated in Figure 5, this deterioration manifests as higher averaged Fiedler value and more failed object localizations.

In previous section, our analysis reveals a strong correlation between the model's localization performance and the averaged Fiedler value of patch feature graphs. This naturally raises the question: can we enhance localization accuracy, and consequently improve MOCL performance, by minimizing the Fiedler value of feature maps' graph Laplacian?

However, direct optimization of the Fiedler value through neural networks is infeasible due to the non-differentiable

**Algorithm 1** CUTER w/ regularization for multi-label online continual learning

---
1: **Input:** Online streaming data $(x, y)$, pre-trained encoder $\theta_0$, classifier $\phi$, model $f = \phi \circ \theta$
2: **Initialize:** Memory buffer $\mathcal{M} \leftarrow \{\}$, Candidate set $\mathcal{C} \leftarrow \{\}$, $\theta \leftarrow \theta_0$
3: **for** $i = 1$ **to** $N$ **do**
4:     Implement MCut to obtain foreground object $x_{obj}$:
5:         Minimize Eq.4
6:         Calculate mask using Eq.6
7:     Select objects into candidate set $\mathcal{C}$ based on:
8:         Prediction $p_{obj} = f(x_{obj})$
9:         Selection threshold (Eq.2)
10: **end for**
11: Compute adjacency matrix $A$ on features $\theta(x)$ and loss using Eq.3
12: Replay memory data $(x^m, y^m)$ according to Eq.1
13: Update model parameters $f$
14: **if** $\mathcal{M}$ is not full **then**
15:     Put candidate set $\mathcal{C}$ in memory buffer $\mathcal{M}$
16: **end if**
17: **if** $\mathcal{M}$ is full **then**
18:     Update memory buffer $\mathcal{M}$ by sampling from $\mathcal{C}$ with class-balanced probabilities
19: **end if**

---

nature of eigenvalue sorting operations on graph Laplacian matrices. Inspired by spectral perturbation theory for symmetric matrices, we approach this challenge from a different perspective. For effective object localization, an ideal feature map should exhibit high intra-object feature similarity while maintaining low inter-object correlations. This property can be mathematically represented by an adjacency matrix $A$ that approximates a symmetric block diagonal structure, where the rank $k$ corresponds to the number of distinct objects in the input image $x$. Such structured adjacency matrices are theoretically guaranteed to yield optimal bipartitions under NCut or MCut criteria.

Without loss of generality, we can assume that during training, the adjacency matrix of each feature graph can be decomposed as $A = A^* + \epsilon$, where $A^*$ represents an ideal block diagonal matrix and $\epsilon$ denotes a non-sparse noise matrix. Based on these assumptions, we can prove that the Fiedler value of the feature graph is directly upper bounded by the norm of this perturbation matrix $\epsilon$.

**Theorem 2.3.** *For any adjacency matrix $A$ can be written as the sum of a block diagonal matrix $A^*$ and a noise matrix $\epsilon$ with zero diagonal entries, the second smallest eigenvalue (Fiedler value) of $A$'s Laplacian matrix $L$ satisfies:*

$$\lambda_2(L) \leq \|\epsilon\|_2 + \|\epsilon\|_\infty$$

Theorem 2.3 provides us a calculable and differentiable

*Table 1.* Comparison results on PASCAL VOC dataset with memory size being $1000 \times 224 \times 224 \times 3$.

| Method | Source Task | Average Performance | | | Last Performance | | |
|---|---|---|---|---|---|---|---|
| | | Avg mAP | Avg CF1 | Avg OF1 | mAP | CF1 | OF1 |
| RS | OCL | 75.05±1.28 | 60.32±1.35 | 61.29±0.74 | 59.71±1.75 | 39.88±1.04 | 46.09±1.24 |
| GSS | | 76.01±1.24 | 60.84±0.82 | 62.37±1.14 | 58.64±1.29 | 40.10±0.82 | 45.13±0.90 |
| iCarl | | 72.67±3.14 | 56.24±2.31 | 58.46±1.78 | 50.77±2.18 | 36.51±1.70 | 41.02±1.91 |
| NsCE | | 75.24±1.41 | 64.42±1.05 | 65.30±0.97 | 56.87±1.14 | 40.78±1.22 | 42.39±1.50 |
| KRT* | MLCIL | 59.45±1.17 | 46.49±0.88 | 57.12±1.38 | 38.90±0.77 | 33.47±1.14 | 36.82±0.95 |
| APPLE | | 76.24±1.25 | 67.01±2.12 | 66.74±1.23 | 58.27±0.82 | 43.77±1.03 | 50.21±0.98 |
| PRS | MOCL | 75.87±0.82 | 58.15±1.46 | 61.62±1.69 | 54.67±0.64 | 42.15±1.15 | 43.87±0.86 |
| OCDM | | 76.14±1.14 | 52.84±0.79 | 65.08±1.11 | 45.35±1.12 | 36.41±0.25 | 40.45±1.01 |
| AGCN | | 75.06±1.01 | 62.37±0.89 | 61.87±2.04 | 57.21±0.74 | 42.06±1.47 | 44.79±1.91 |
| AGCN++ | | 74.14±0.78 | 65.04±1.24 | 63.55±1.45 | 55.34±0.87 | 40.06±0.75 | 44.21±1.14 |
| CUTER w/ $R_l$ | | **82.07±0.53** | **72.19±0.70** | **75.27±0.57** | **67.89±1.28** | **51.35±0.98** | **59.98±1.17** |

alternative to enhance the model's localization capacity. Although during MOCL training we cannot obtain the ideal adjacency matrix $A^*$ for each sample's features to directly constrain the corresponding noise matrix $\epsilon$, considering that we hope to guide the adjacency matrix $A$ toward an ideal block diagonal structure, directly imposing constraints on $A$ is also a viable choice, as shown in Eq.3.

$$L = L_{asy}(f, x, y) + R(A) \qquad (3)$$

This matrix decomposition perspective relates to robust graph structure learning through noise suppression. Drawing from common regularization terms in graph learning (low-rank, sparsity, and smoothness), we find the nuclear norm $R(A) = ||A||_*$ particularly effective for reducing the noise matrix $\epsilon$ in MOCL (Figure 5). Detailed comparisons and more analysis are presented in Table 3 and Appendix D.

Up till now, we have fully presented our proposed cut-out-and-experience-replay strategy (CUTER) for MOCL with a detailed algorithm procedure in **Algorithm 1**. For proofs(Appendix C.2), detailed explanations (Appendix D.2), please refer to Appendix.

## 3. Experiments

### 3.1. Datasets and Experimental Setting

**Datasets.** Following the experimental frameworks of previous works(Dong et al., 2023; Liang & Li, 2022), we evaluate our method on three benchmark datasets: MS-COCO 2014(Lin et al., 2014), PASCAL VOC 2007(Everingham et al., 2015), and NUS-WIDE(Chua et al., 2009). Specifically, PASCAL VOC 2007 consists of 5,011 training and 4,952 test images spanning 20 classes, averaging 2.4 labels per image. MS-COCO contains 82,081 training and 40,504 validation images across 80 classes, with an average of 2.9 labels per image. As NUS-WIDE is no longer publicly available, we reconstructed the dataset by re-scraping from Flickr, obtaining 126,034 training and 84,226 test images across 81 classes, with an average of 2.4 labels per image.

**Experiments Setup.** Following prior studies (Mai et al., 2021; Liang & Li, 2022; Dong et al., 2023), we partition the datasets into several tasks with disjoint given label sets but overlapping ground truth support sets, simulating a realistic multi-label data stream. Unlike some multi-label class incremental studies that rely on a base task, we argue that with pre-trained models, this online learner should be capable of rapid adaptation to dynamic data streams from any starting point. Thus, we divide PASCAL VOC 2007 into 5 tasks with 4 classes each, MS-COCO into 8 tasks with 10 classes each, and NUS-WIDE into 8 tasks where the first task has 11 classes and the remaining tasks have 10 classes each. For all datasets, the order of class assignments to tasks follows the lexicographical order of class names, as described in (Dong et al., 2023; Du et al., 2024; 2025).

**Baseline Methods.** In our experiments, we compare CUTER with 10 advanced continual learning methods spanning OCL, multi-label class incremental learning (MLCIL), and MOCL. For comprehensive evaluation, we adapt four OCL methods to our MOCL setting by modifying their classifier architectures and classification loss functions, including representative replay-based methods RS(Chaudhry et al., 2019b), GSS(Aljundi et al., 2019b), iCarl(Rebuffi et al., 2017), and the recent comprehensive approach NsCE(Xinrui et al., 2024). We also evaluate six multi-label continual learning methods across MLCIL and MOCL categories. Among these, KRT(Dong et al., 2023), APPLE(Song et al., 2024), AGCN and AGCN++(Du et al., 2022; 2023) primarily address missing labels through knowledge distillation or label correlation, while PRS(Kim et al., 2020) and OCDM(Liang & Li, 2022) focus on class imbalance via carefully designed sampling strategies. Additional details on each method are provided in the Appendix E.1.

**Evaluation Metrics.** Following (Du et al., 2023; 2024), we use several standard metrics: mean average precision (mAP), per-class F1 score (CF1), and overall F1 score (OF1). Beyond conventional multi-label classification evaluation, we report both cross-task average performance (on seen classes) and model's final performance on all classes $\mathcal{C}^{1:K}$.

*Table 2.* Comparison results on MSCOCO dataset with memory size being $1000 \times 224 \times 224 \times 3$.

| Method | Source Task | Average Performance | | | Last Performance | | |
|---|---|---|---|---|---|---|---|
| | | Avg mAP | Avg CF1 | Avg OF1 | mAP | CF1 | OF1 |
| RS | | 48.12±1.78 | 32.21±1.04 | 37.36±0.92 | 18.97±2.01 | 6.44±1.56 | 12.84±1.05 |
| GSS | OCL | 49.41±0.84 | 32.85±0.70 | 38.26±0.55 | 25.67±1.34 | 11.82±0.96 | 17.40±1.18 |
| iCarl | | 47.51±1.30 | 33.09±0.86 | 37.51±1.09 | 19.24±1.37 | 7.62±1.38 | 12.51±0.89 |
| NsCE | | 51.24±0.95 | 37.10±1.10 | 44.73±0.96 | 26.19±1.24 | 17.07±1.04 | 22.30±0.94 |
| KRT* | MLCIL | 44.34±0.95 | 39.06±1.08 | 43.51±1.17 | 36.51±0.87 | 27.41±0.92 | 30.16±1.15 |
| APPLE | | 48.67±1.28 | 42.34±0.85 | 46.86±1.31 | 38.47±0.94 | 30.12±1.08 | 33.51±0.67 |
| PRS | | 51.79±1.08 | 33.64±1.32 | 38.06±0.94 | 27.95±1.98 | 15.33±2.47 | 18.22±1.29 |
| OCDM | MOCL | 55.45±0.87 | 46.78±1.02 | 50.59±0.92 | 40.56±0.80 | 28.45±0.71 | 31.29±0.52 |
| AGCN | | 56.45±0.92 | 48.03±1.35 | 51.27±0.91 | 37.48±0.73 | 27.82±0.96 | 32.38±1.29 |
| AGCN++ | | 54.31±0.82 | 47.69±0.47 | 52.27±0.85 | 36.45±0.74 | 29.64±0.83 | 33.05±0.71 |
| CUTER w/ $R_l$ | | **60.14±0.60** | **51.53±0.61** | **54.92±0.62** | **47.82±0.60** | **35.94±0.71** | **39.18±0.65** |

**Implementation Details.** After evaluating the localization capacity of different pre-trained models, we adopt ImageNet-21k pre-trained (DINO v1) ViT-S/16 as our backbone. All methods compared in Tab 1, Tab 2 and Tab 4 use the same backbone and pre-trained initialization, except KRT which uses TresnetM, as our attempts to reproduce KRT with ViT-S/16 did not yield superior performance to the original TresnetM implementation. We follow the data augmentation strategy from (Dong et al., 2023). All experiments in the main text were repeated five times, and we report the mean and standard deviation. Additional implementation details are provided in the Appendix E.1.

### 3.2. Overall Performance

At first, we conduct a comprehensive evaluation of our proposed CUTER by comparing its performance with several existing state-of-the-art MOCL methods and various continual learning variants. Our comparison includes online continual methods, multi-label class incremental learning methods (MLCIL), and MOCL methods. While there are more recent MLCIL methods like CSC and RebLL(Du et al., 2024; 2025), we excluded them from our comparison due to their unavailable implementation details (no public code links) and different experimental settings. Tables 1, 2, and 4 display both average and overall performance across three synthetic benchmark datasets. The results demonstrate that our proposed method, CUTER, consistently outperforms other approaches. Notably, CUTER shows particularly significant performance improvements in the later tasks of continual learning, with more pronounced advantages in terms of last performance on all classes.

Secondly, we compare three commonly used regularization in robust graph structure learning as additional loss terms for localization capacity consolidation. Specifically, we implement low rank regularization $R_l = \|A\|_*$, sparse regularization $R_{sp} = \|A\|_{l1}/\|A\|_{l2}$, where $\|A\|_{l1}$ and $\|A\|_{l2}$ represent the $l1$ and $l2$ norms calculated after flattening matrix $A$ into a vector, following the definition and constraints of sparsity from (Xinrui et al., 2024), and smooth regular-

ization $R_{sm} = \frac{1}{2} \sum_{i,j=1}^{N} A_{ij}(\theta(x_i) - \theta(x_j))^2$, where $\theta(x_i)$ represents the feature corresponding to the $i$-th patch of the input image. As demonstrated in Table 3, compared with $R_l$ which can consistently boost model's performance, $R_{sm}$ and $R_{sp}$ often lead to performance degradation. We hypothesize that this inconsistency might stem from potential conflicts between these so-called ideal graph structures and the model's classification objectives, which we discuss in detail in Appendix D.

Thirdly, we evaluate CUTER as a plug-in component for other popular technique in methods including PRS, OCDM, KRT, and AGCN. For fair comparison, we use a basic version (Cut.Rep) without re-balanced sampling and low-rank regularization. As shown in Table 5, our strategy effectively complements different re-balanced sampling methods, and can be further enhanced by techniques like knowledge distillation and graph-based label mining.

Additional visualization and experimental results, including mAP curves throughout the training, performance across different backbone architectures (pre-trained initializations), and throughput evaluations of different methods, can be found in Appendix E.2.

### 3.3. Ablation Studies

In this section, we conduct a series of ablation studies to investigate the specific effects of different proposed components. From Table 5 and Table 6, we can draw several conclusions: (1) Each component (cut and replay, re-balanced sampling, and low rank regularization) provides consistent performance improvements, with cut and replay (Cut.Rep) showing the most significant effect. (2) While our proposed re-balanced sampling method may not consistently outperform other carefully designed sampling methods like PRS and OCDM across all datasets and settings, it achieves comparable results with a lower computational cost ($O(B)$ compared with $O(B|\mathcal{M}|)$ in OCDM, where $B$ is the batch size). (3) Compared with a fixed backbone, updating the model continuously during training improves performance,

*Table 3.* Comparison of different regularization terms for preserving localization ability.

| Model | Cut.Rep | $R_l$ | $R_{sp}$ | $R_{sm}$ | VOC | | COCO | | NUSWIDE | |
| --- | --- | --- | --- | --- | --- | --- | --- | --- | --- | --- |
| | | | | | Avg mAP | Last mAP | Avg mAP | Last mAP | Avg mAP | Last mAP |
| CUTER w/ $R_{sp}$ | ✓ | | ✓ | | 78.34±0.76 | 66.27±0.85 | 58.68±1.03 | 46.74±0.62 | 49.73±0.50 | 37.06±1.16 |
| CUTER w/ $R_{sm}$ | ✓ | | | ✓ | 79.01±0.84 | 65.63±0.92 | 58.31±0.68 | 44.90±0.85 | 50.14±0.76 | **38.09±1.34** |
| CUTER w/ $R_l$ | ✓ | ✓ | | | **82.07±0.53** | **67.89±1.28** | **60.14±0.60** | **47.82±0.60** | **51.14±0.72** | 37.17±1.46 |

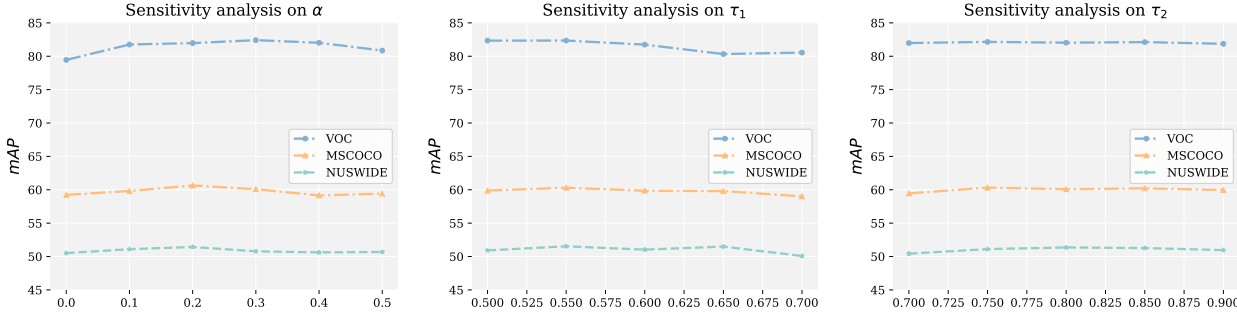

*Figure 6.* Sensitivity analysis on coefficient $\alpha$ and confidence thresholds $\tau_1$ and $\tau_2$.

*Table 4.* Comparison results on NUSWIDE dataset with memory size being $1000 \times 224 \times 224 \times 3$.

| Method | Avg mAP | Avg CF1 | Last mAP |
| --- | --- | --- | --- |
| RS | 43.90±1.08 | 36.79±0.76 | 26.78±0.59 |
| GSS | 44.28±0.76 | 37.01±1.14 | 28.22±0.90 |
| iCarl | 43.50±0.83 | 35.74±0.95 | 27.60±0.85 |
| NsCE | 45.72±0.52 | 34.95±0.75 | 27.14±0.60 |
| KRT* | 47.30±1.07 | 39.70±1.24 | 31.25±1.19 |
| APPLE | 47.53±0.76 | 40.89±0.97 | 32.40±1.34 |
| PRS | 42.74±0.84 | 35.01±0.92 | 22.78±0.72 |
| AGCN | 49.16±0.97 | 38.41±1.05 | 33.49±0.88 |
| AGCN++ | 49.03±1.35 | 40.17±1.28 | 32.09±1.41 |
| OCDM | 40.05±0.42 | 33.66±0.87 | 29.41±0.65 |
| CUTER w/ $R_l$ | **51.14±0.72** | **42.92±1.03** | **37.57±1.46** |

*Table 5.* Cut and Replay (CutRep) works as a plug-in component for several MOCL methods (Averaged mAP reported).

| Method | VOC | MSCOCO | NUSWIDE |
| --- | --- | --- | --- |
| Cut.Rep | 77.92±0.78 | 53.40±0.82 | 46.30±1.02 |
| Cut.Rep w/ PRS | **81.35±1.02** | 58.72±0.94 | 50.69±0.45 |
| Cut.Rep w/ OCDM | 81.09±0.67 | 57.90±0.73 | **51.21±1.38** |
| CUTER | 79.45±0.92 | 59.23±0.72 | 50.51±0.64 |
| CUTER w/ KRT | 79.37±1.25 | 57.09±1.32 | 50.56±0.95 |
| CUTER w/ AGCN | 76.95±1.14 | **59.31±0.94** | 48.95±0.89 |

as it naturally leads to better localization and cut-out quality on the current task.

### 3.4. Sensitivity Analysis

In this section, we analyze the impact of coefficient $\alpha$ of the low rank regularization term and the thresholds $\tau_1$ and $\tau_2$ on foreground object selection for memory buffer in experience replay. As shown in Figure 6, our proposed method has demonstrated relatively robust outcomes when $\alpha, \tau_1, \tau_2$ remain within certain ranges. It should be noted that co-

efficient $\alpha$ shouldn't be too large since in our method this low rank regularization is implemented without a common constraint to regulate the magnitude of changes in adjacency matrix $A$. Therefore, an excessively big $\alpha$ would cause $A$ to approach a zero matrix, which is obviously undesirable. Meanwhile, an excessively large $\tau_1$ would make it difficult for the model to select a sufficient number of tail-class samples, thereby affecting the overall performance.

## 4. Conclusion

In this study, we highlight the fundamental question behind two key challenges (pervasive missing labels and uncontrollable class imbalance) that existing MOCL methods strive to tackle. By exploring and leveraging the innate localization capacity of widely used pre-trained models, we propose a method called CUTER to comprehensively address the existing limitations of MOCL methods through identification and replay of regions corresponding to different given labels. Additionally, we further explore how to enhance model localization (unsupervised detection and segmentation) performance in such an online incremental learning framework from the perspective of graph spectral theory. Our method not only achieves a remarkable performance boost but also differs significantly from existing MOCL approaches. We hope this study can bring fresh perspectives to the area of multi-label learning in this continual setting.

## Impact Statement

This research aims to advance the field of Machine Learning. While our work has potential societal implications, we believe discussing specific scenarios is beyond the scope of

*Table 6.* Ablation studies. We denote low rank regularization as $R_l$, sparse regularization as $R_{sp}$, and smooth regularization as $R_{sm}$. CUTER$^-$ refers to CUTER using a fixed backbone for object localization and the later cut out operation.

| Model | Rep | Cut.Rep | Re-balance | $R_l$ | VOC | | COCO | | NUSWIDE | |
|---|---|---|---|---|---|---|---|---|---|---|
| | | | | | Avg mAP | Last mAP | Avg mAP | Last mAP | Avg mAP | Last mAP |
| Baseline(RS) | ✓ | | | | 75.05±1.28 | 59.71±1.75 | 48.12±1.78 | 18.97±2.01 | 43.90±1.08 | 26.78±0.59 |
| Cut.Rep | | ✓ | | | 77.92±0.78 | 64.52±1.07 | 53.40±0.82 | 31.17±0.69 | 46.30±1.02 | 30.41±0.73 |
| Cut.Rep w/ $R_l$ | | ✓ | | ✓ | 78.24±0.93 | 63.95±0.76 | 56.37±0.64 | 33.84±1.01 | 48.12±0.85 | 31.46±0.71 |
| CUTER$^-$ | | ✓ | ✓ | | 78.62±0.77 | 64.95±0.83 | 59.01±0.45 | 45.19±0.80 | 49.67±0.72 | 36.42±0.91 |
| CUTER | | ✓ | ✓ | | 79.45±0.92 | 66.09±1.03 | 59.23±0.72 | 45.79±0.85 | 50.51±0.64 | **37.35±0.42** |
| CUTER w/ $R_l$ | | ✓ | ✓ | ✓ | **82.07±0.53** | **67.89±1.28** | **60.14±0.60** | **47.82±0.60** | **51.14±0.72** | 37.17±1.46 |

this technical paper.

## Acknowledgements

This work is supported by National Science and Technology Major Project (2022ZD0114801), National Natural Science Foundation of China (No. 62376126), Funding for Outstanding Doctoral Dissertation in NUAA (BCXJ25-21) and Major Special Basic Research Project for Aero-engines and Gas Turbines: Research on Fault Diagnosis and Prediction Technology for Typical Transmission Components (J2019-IV-0018-0086). Additionally, we thank the anonymous reviewers for their valuable comments and suggestions.

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

## A. Detailed Related Works

### A.1. Online Continual Learning

OCL serves as a more realistic extension of continual learning, addressing scenarios where data distributions dynamically change over time, in contrast to traditional batch learning where complete task datasets are available upfront(Gunasekara et al., 2023). To mitigate catastrophic forgetting, most OCL methods maintain a memory buffer for experience replay, employing various strategies for memory management and sample selection(Chaudhry et al., 2019a;b; Aljundi et al., 2019a;b; Chaudhry et al., 2020; Shim et al., 2021; Wang et al., 2022; Caccia et al., 2022; Ghunaim et al., 2023; Xinrui et al., 2024). Additionally, researchers have explored approaches to enhance feature learning and classifier adaptation in single-pass training scenarios(Rebuffi et al., 2017). Recent advances incorporate techniques such as contrastive learning(Mai et al., 2021; Cha et al., 2021), mutual information maximization(Gu et al., 2022; Guo et al., 2022), and prototype learning(Zhu et al., 2021; Wei et al., 2023) to improve model discriminative power and overall performance.

### A.2. Multi-label Classification

Multi-label learning has long been a fundamental paradigm in machine learning(Zhang & Zhou, 2013; Wei & Li, 2019b; Li et al., 2024), attracting substantial research attention. In this review, we focus on works that extract label-specific features or regions, which are closely related to our approach. These studies can be broadly categorized into two groups.

The first category decomposes multi-label classification into binary (one-vs-rest) subproblems, where each label is mapped to a distinct feature subset. This category encompasses three main approaches: (1) Clustering-based methods (Zhang & Wu, 2014; Ren et al., 2019) partition the feature space into label-specific clusters by exploiting both independent and co-occurrence patterns between features and labels; (2) Label correlation mining approaches (Huang et al., 2015; 2016; Hang & Zhang, 2022) discover and utilize the interdependencies among labels by constructing label correlation matrix through a specially designed module like graph encoder(Hang & Zhang, 2021) to guide feature selection, typically through matrix factorization or graph-based techniques; (3) Regularized feature learning methods (Wei et al., 2018; Li et al., 2022; 2025) impose structural constraints on the feature selection process, often using $\ell_1$ or group sparsity regularizers to encourage label-specific feature sparsity. However, since these methods operate on abstract feature vectors rather than spatial representations, they cannot support direct sample-level operations like cut-out augmentation.

The second category adopts a more direct approach by leveraging visual attention mechanisms to localize label-specific regions (Chen et al., 2019; Narayan et al., 2021; Li et al., 2023). Specifically, these methods either utilize the inherent attention mechanisms in deep networks, employ visualization techniques like Grad-CAM (Selvaraju et al., 2017), or incorporate dedicated attention modules (Wang et al., 2017; Gao & Zhou, 2021). While these approaches achieve coarse-grained localization of label-relevant regions, their attention maps often focus on local discriminative features rather than complete object regions. This limitation makes them less suitable for MOCL, which requires precise object boundaries and comprehensive object representations.

Noting the extensive approaches proposed for label-specific feature learning in traditional multi-label learning(Yu & Zhang, 2021; Hang & Zhang, 2021; Hang et al., 2022; Hang & Zhang, 2022; Chen et al., 2019; Narayan et al., 2021; Li et al., 2023), we emphasize that they rely on complete label set with heavy computation overload, or fail to identify the label specific regions, which are inherently defected by the high-efficiency requirement of MOCL task encountering the missing label factor.

### A.3. Multi-label Online Continual Learning

Multi-label online continual learning or multi-label class incremental learning emerge as a novel research field that integrates the advantages of continual learning and multi-label learning(Zhang & Zhou, 2007; 2013; Zhang & Wu, 2014; Zhang et al., 2020; Wei et al., 2021). Beyond the widely discussed catastrophic forgetting, existing MOCL studies primarily address two key challenges: (1) the missing past and future labels inherent to the intersection of continual learning and multi-label classification; (2) the intrinsic and uncontrollable class imbalance prevalent in conventional multi-label data streams. Researchers have addressed missing label challenges by leveraging stored historical models through techniques proven effective in multi-label learning and continual learning, such as GCN(Du et al., 2022; 2023; 2025), knowledge distillation(Dong et al., 2023) and vision language model(Zhao et al., 2025). The class imbalance problem has been tackled through carefully designed sampling strategies for replay methods. Notable examples include PRS(Kim et al., 2020), which employs heuristic sampling, and OCDM(Liang & Li, 2022), which utilizes optimization-based sampling, both aiming to

mitigate the impact of long-tailed class distributions in continual multi-label scenarios. However, these existing methods overlook the root cause of these challenges and a crucial issue in multi-label learning(Zhao et al., 2016; Wang et al., 2017; Gao & Zhou, 2021): how to identify and discriminate label-specific regions and their corresponding features in an online manner? Our work directly addresses this fundamental question.

## B. Preliminaries

In this section, we first review two unsupervised segmentation techniques: Normalized Cut(Shi & Malik, 2000) and Mask Cut(Wang et al., 2023a), which motivate our investigation into label-specific region identification in MOCL.

Normalized cut (NCut) formulates image segmentation as a graph partitioning problem. Let $G = (V, E)$ be a weighted undirected graph, where $V$ denotes the vertex set with each vertex $v$ corresponding to an image patch (feature patch), $E$ represents the edge set, and $W$ is the edge weight matrix with entries $W_{ij}$ encoding the similarity between vertices $i$ and $j$. NCut aims to partition the graph into disjoint sets $A$ (background) and $B$ (foreground object $b$) by minimizing the NCut energy $\mathcal{E}(A, B, V)$:

$$\mathcal{E}(A, B, V) = \frac{\mathcal{C}(A, B)}{\mathcal{C}(A, V)} + \frac{\mathcal{C}(A, B)}{\mathcal{C}(B, V)}, \tag{4}$$

where $\mathcal{C}$ measures the similarity between two sets, e.g., $\mathcal{C}(A, B) = \sum_{v_i \in A, v_j \in B} W_{ij}$.

Meanwhile, mask cut (MCut) extends NCut to identify multiple objects per image by iteratively applying NCut to a masked similarity matrix. At the $t$-th iteration, after obtaining the bipartition from NCut, the algorithm divides patches into two disjoint groups (background $a^t$ and object $b^t$) and constructs a binary mask $M^t$, where

$$M_{ij}^t = \begin{cases} 1, & \text{if } M_{ij} > \text{mean}(b^t) \\ 0, & \text{otherwise} \end{cases} \tag{5}$$

Upon determining the foreground group $b^t$, MCut updates the node similarity $W_{ij}^{t+1}$ by masking out nodes corresponding to the foreground from previous stages $t$:

$$W_{ij}^{t+1} = \frac{(v_i \prod_{s=1}^t \hat{M}_{ij}^s)(v_j \prod_{s=1}^t \hat{M}_{ij}^s)}{||v_i||_2 ||v_j||_2} \tag{6}$$

where $\hat{M}_{ij}^s = 1 - M_{ij}^s$. With the updated $W_{ij}^{t+1}$, MCut iteratively minimizes Eq.4 and calculates Eq.6 to generate multiple segmentation masks for distinct objects.

## C. Theoretical Foundations of CUTER

### C.1. Relationship Between Normalized Cut and Graph Spectral Theory

As discussed in Section B, the Normalized Cut (NCut) can be formulated as a graph partitioning problem. Given a graph $G = (V, E)$ with vertex set $V$, edge set $E$, and edge weights $W_{ij}$ representing the similarity between nodes $i$ and $j$, the objective is to partition the graph into disjoint subsets $A$ and $B$ that: (1) minimize the inter-subset similarity, and (2) maximize the intra-subset similarity. The NCut objective function is defined as:

$$\frac{\mathcal{C}(A, B)}{\mathcal{C}(A, V)} + \frac{\mathcal{C}(A, B)}{\mathcal{C}(B, V)}$$

where $\mathcal{C}$ measures the similarity between two sets: $\mathcal{C}(A, B) = \sum_{v_i \in A, v_j \in B} W_{ij}$.

The NCut problem is NP-hard due to its discrete nature. To make it computationally tractable, it is relaxed into a continuous optimization problem using spectral graph theory. This involves the graph Laplacian $L = D - W$. To prevent nodes with extremely large degrees from dominating the graph structure, the **symmetric normalized Laplacian** $L_{\text{sym}}$ is employed:

$$L_{\text{sym}} = I - D^{-1/2} W D^{-1/2} \tag{7}$$

The problem can then be reformulated as minimizing $y^\top L_{\text{sym}} y$ subject to constraints ensuring $y$ is balanced and orthogonal to the trivial solution (i.e., $y^\top D\mathbf{1} = 0$, where $\mathbf{1}$ is the all-ones vector). After relaxing $y$ to take continuous values, the

problem becomes:

$$\min_{y} y^\top L_{\text{sym}} y$$

subject to:

$$y^\top D\mathbf{1} = 0, \quad y^\top Dy = 1$$

The solution is the eigenvector corresponding to the **second smallest eigenvalue** of $L_{\text{sym}}$. This continuous solution can be discretized through thresholding to obtain the final partition. For a complete proof, refer to (Shi & Malik, 2000).

### C.2. Proof of Theorem 2.3

In this section, we first give a detailed description of the adjacency matrix of each feature graph. The features of input data can be divided into different patches, by using functions like Gaussian kernel to construct an undirected graph, with a $N \times N$ adjacency matrix $A$. Then, without loss of generality, we can make the following assumption:

**Assumption C.1.** The adjacency matrix $A$ of each feature graph can be decomposed as an ideal block diagonal matrix $A^*$ and a non-sparse and potentially high rank noise matrix $\epsilon$:

$$A = A^* + \epsilon$$

With Assumption C.1, we can rewrite the graph Laplacian $L$ as:

$$L = D - A = D^* + \Delta D - (A^* + \epsilon)$$

where $\Delta D = D - D^*$ with $D^*$ is a $N \times N$ diagonal degree matrix with elements $d(i) = \sum_j A_{ij}^*$. Thus, we can write the elements of $\Delta D$ as:

$$\Delta D_{ii} = \sum_{j=1}^{N}(A_{ij} - A_{ij}^*) = \sum_{j=1}^{N} \epsilon_{ij}.$$

Since it's clear to see that $\Delta D$ is also a diagonal matrix, we have:

$$||\Delta D||_2 = max_i |\Delta D_{ii}| \leq max_i \sum_{j=1}^{N} |\epsilon_{ij}| = ||\epsilon||_\infty.$$

We can conclude that $||\Delta D||_2$ is directly upper bounded by $\epsilon$. Based on this, we can now turn to analyze the upper bound of the disturbation of graph Laplacian. It easy to conclude that:

$$\Delta L = L - (D^* - A^*) = \Delta D - \epsilon.$$

Thus, we have:

$$||\Delta L||_2 \leq ||\Delta D||_2 + ||\epsilon||_2 \leq ||\epsilon||_\infty + ||\epsilon||_2.$$

It means that if we can bound $\lambda_2(L)$ by $||\Delta L||_2$, we can finish this proof. Before doing this part, we first introduce an important Lemma:

**Lemma C.2.** *(Courant-Fischer Formula (Zhang, 1997)) For a Hermitian $A \in \mathbb{C}^{N \times N}$,*

$$\lambda_k = \min_{\dim(S)=n-k+1} \max_{0 \neq x \in S} \frac{x^* A x}{x^* x} = \max_{\dim(S)=k} \min_{0 \neq x \in S} \frac{x^* A x}{x^* x}, \quad k = 1:n.$$

From Lemma C.2, the Courant–Fischer theorem yields the following upper and lower bounds:

$$\lambda_k(A) + \lambda_n(B) \leq \lambda_k(A + B) \leq \lambda_k(A) + \lambda_1(B).$$

from which it follows that

$$\max_k |\lambda_k(A + B) - \lambda_k(A)| \leq \max(|\lambda_n(B)|, |\lambda_1(B)|) = \|B\|_2.$$

This inequality shows that the eigenvalues of a Hermitian matrix are well conditioned under perturbation. We can rewrite the inequality in the symmetric form

$$\max_k |\lambda_k(A) - \lambda_k(B)| \leq \|A - B\|_2.$$

By substituting $L$ for $A$ and $L^* = D^* - A^*$ for $B$ and taking $k = 2$, we have:

$$\lambda_2(L) - \lambda_2(L^*) \leq \|\Delta L\|_2.$$

Then, according to Assumption C.1, we can deduce that $L^*$ is also an ideal block diagonal matrix, therefore its second smallest eigenvalue $\lambda_2(L^*) = 0$. Combined with our previous results, we obtain:

$$\lambda_2(L) \leq \|\Delta L\|_2 \leq \|\epsilon\|_\infty + \|\epsilon\|_2.$$

This completes the proof.

# D. Analysis on Models' Localization Ability

## D.1. Pre-trained Model

In this section, we discuss pre-trained models' localization capabilities and their underlying mechanisms. Based on the experimental results shown in Figure 2 and Table 7, we observe a clear hierarchy in terms of zero-shot object localization ability and potential for MOCL: contrastive learning models with multi-crop augmentation (e.g., DINO v1, DINO v2) demonstrate superior localization capabilities compared to standard contrastive learning approaches (e.g., MoCo, SimCLR), which in turn outperform conventional supervised training methods. Reconstruction-based methods like MAE and iBOT exhibit relatively weaker performance in this aspect. We exclude CUTLearn from this comparison as it is a specialized model designed specifically for zero-shot detection that requires additional fine-tuning, making it fundamentally different from the aforementioned general-purpose pre-trained models.

When examined through the lens of spectral graph theory, these results become more intuitive. The consistency between augmented views employed in contrastive learning naturally guides the model to focus on the most salient objects in images. This effect is particularly pronounced because pre-training datasets like ImageNet are typically single-labeled and object-centered, making multi-crop augmentation consistency especially effective. This training dynamic implicitly encourages feature similarity among patches belonging to the same object, thereby enhancing the separability of the feature-constructed graph (metric like cheeger constant). In contrast, supervised learning demonstrates weaker performance in this aspect, though it still retains some localization capability since images from the same class typically contain similar objects, leading to comparable feature representations even without explicit intra-image contrasting. However, mask reconstruction pre-training methods, despite their proven effectiveness in feature learning for various downstream tasks, show the poorest performance in this aspect. From a feature graph perspective, the indiscriminate random masking and reconstruction likely diminishes the feature coherence between different patches of the same object, which adversely affects direct feature-based zero-shot localization or segmentation capabilities.

## D.2. Further Analysis on Low-rank Regularization

Constraining the nuclear norm of a matrix $A$ (i.e., minimizing $\|A\|_*$, the sum of singular values of $A$) has a significant impact on reducing both the spectral norm $\|\Delta A\|_2$ and the infinity norm $\|\Delta A\|_\infty$, where $\Delta A = \epsilon = A - A^*$. Below, we present a detailed explanation.

**Reduction in Spectral Norm** $\|\Delta A\|_2$: Minimizing $\|A\|_*$ applies *soft-thresholding* to the singular values of $A$, reducing the rank of $A$ and suppressing the singular values of $\Delta A$:

$$\|\Delta A\|_2 \leq \|\Delta A\|_*.$$

**Reduction in Infinity Norm** $\|\Delta A\|_\infty$: Low-rank matrices have limited row variability, which constrains $\|\Delta A\|_\infty$. Since:

$$\|\Delta A\|_\infty \leq \sqrt{n}\|\Delta A\|_2,$$

reducing $\|\Delta A\|_2$ indirectly reduces $\|\Delta A\|_\infty$.

Similarly, sparse regularization shows comparable effects in de-noising $\epsilon$. However, unlike low-rank regularization, it disrupts the inherent structural properties of ViT parameters (Darcet et al., 2023), potentially compromising classification capacity. This may explain its inferior performance in MOCL, as shown in Table 3. As for smooth regularization, we argue it may not benefit localization ability since overly similar node features could hinder spectral clustering-like operations.

## E. Additional Experiments and Implementation Details

### E.1. Implementation Details

In this part, we provide implementation details and configurations of the methods compared in Table 1, Table 2, and Table 4:

For RS(Chaudhry et al., 2019b), GSS(Aljundi et al., 2019b), iCarl, and NsCE(Xinrui et al., 2024), we adopted their original codebases with minor modifications to their classifier architectures and classification loss functions. For methods employing prototypical classifiers (iCarl and NsCE), we computed cosine similarities and applied a threshold of 0.5 to determine positive labels. Regarding the classification loss, we replaced the conventional cross-entropy loss designed for single-label scenarios with the asymmetric loss widely adopted in multi-label classification settings.

For PRS(Kim et al., 2020), we directly used their provided code[1] with their reported optimal hyper-parameter $\rho = 0.5$. As for OCDM(Liang & Li, 2022), since their code was not publicly available but shares similar principles with PRS, we implemented it using the same hyper-parameter $\rho = 0.5$ and report our reproduction results.

For KRT and APPLE, both designed for Multi-label Class Incremental Learning, we ensured fair comparison by implementing vanilla experience replay based on the official KRT codebase[2], with each method trained for one epoch.

For AGCN and AGCN++, we utilized the official implementation of AGCN[3] with their reported optimal hyper-parameters. Since AGCN++(Liang & Li, 2022) code was not publicly available but follows similar principles to AGCN, we implemented it using the same hyper-parameters and report our reproduction results.

For other common hyperparameters such as learning rate, weight decay, and batch size, we conducted a grid search and report the best results for all compared methods. Unless otherwise specified above, we consistently used the AdamW optimizer with a learning rate of 1e-4, and a weight decay of 1e-4. For PASCAL VOC, we set the data stream batch size as 12, memory buffer sampling batch size of 6. For COCO and NUSWIDE, we set the data stream batch size as 20, memory buffer sampling batch size of 5.

### E.2. Additional Experiments

We first provide a detailed visualization of different methods' performance (mAP%) across various datasets. As shown in Figure 7, our proposed CUTER consistently demonstrates superior performance throughout the entire training process. We observe that MLCIL methods (e.g., APPLE and AGCN++) exhibit significantly degraded performance compared to their offline implementations, especially in the early training stages. This performance gap highlights a key distinction between MOCL and MLCIL: methods relying on iterative pseudo-labeling struggle to adapt to high-velocity data streams.

Secondly, we investigate how backbone architectures and pre-training strategies affect the downstream performance of MOCL, as shown in Table 7. Our method relies more heavily on backbones with strong localization capabilities and their corresponding pre-training approaches. As discussed in Appendix D, DINO pre-trained ViT models demonstrate superior performance. We also reproduced OCDM, a current SOTA MOCL method, for comparative analysis. The results show that our method maintains its advantages when using ViT as the backbone, while performance with ResNet backbones is relatively weaker. This discrepancy may be attributed to our treatment of ResNet feature maps - although we divided them into patches similar to ViT's approach and applied MCut for foreground object detection and cropping, this adaptation may not be optimal for CNN-based architectures.

Thirdly, we analyze the model throughput of our approach compared to state-of-the-art methods. As shown in Figure 8, while vanilla experience replay (RS) achieves the highest model throughput, it demonstrates relatively weaker performance. Our method achieves superior results at the cost of increased computational overhead, primarily due to the multi-round MCut operations that cannot be parallelized on GPUs. Additionally, computing the rank approximation (nuclear norm) for

---

[1] https://github.com/cdjkim/PRS
[2] https://github.com/witdsl/KRT-MLCIL
[3] https://github.com/Kaile-Du/AGCN

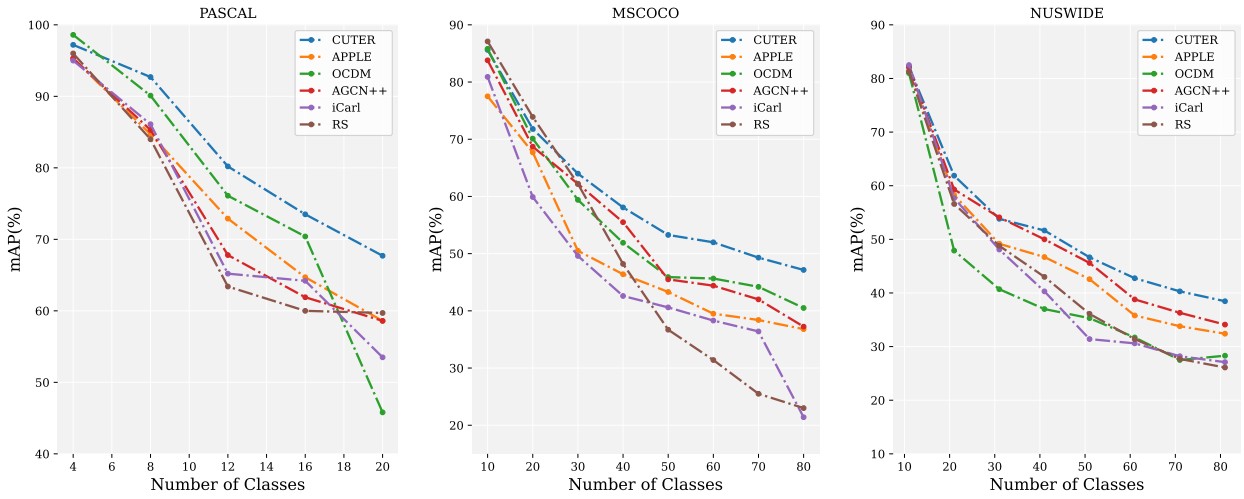

*Figure 7.* Overall training results on PASCAL VOC, MSCOCO and NUSWIDE. We include 5 different methods across the OCL, MLCIL and MOCL methods.

*Table 7.* Performance comparison across different backbone architectures and pre-training methods.

| Model | Backbone | Pre-training Method | VOC | | COCO | | NUSWIDE | |
|---|---|---|---|---|---|---|---|---|
| | | | Avg mAP | Last mAP | Avg mAP | Last mAP | Avg mAP | Last mAP |
| CUTER w/ $R_l$ | Vit-S | Supervised | 79.56 | 63.82 | 57.92 | 43.04 | 48.36 | 35.42 |
| OCDM | Vit-S | Supervised | 75.35 | 44.21 | 53.26 | 39.84 | 41.05 | 26.87 |
| CUTER w/ $R_l$ | Vit-S | MoCo v3 | 80.47 | 64.92 | 58.10 | 45.72 | 50.29 | 34.70 |
| OCDM | Vit-S | MoCo v3 | 75.94 | 46.32 | 56.14 | 42.75 | 40.85 | 30.01 |
| CUTER w/ $R_l$ | Vit-T | DINO v1 | 81.84 | 67.50 | 59.38 | 47.95 | 52.08 | 36.68 |
| CUTER w/ $R_l$ | Vit-S | DINO v1 | 82.07 | 67.89 | 60.14 | 47.82 | 51.14 | 37.47 |
| CUTER w/ $R_l$ | Vit-B | DINO v1 | 83.25 | 68.14 | 61.08 | 45.70 | 50.91 | 37.52 |
| OCDM | Vit-S | DINO v1 | 76.14 | 45.35 | 55.45 | 40.56 | 40.37 | 29.41 |
| CUTER w/ $R_l$ | Vit-S | MAE | 77.31 | 60.45 | 56.42 | 44.06 | 47.24 | 32.40 |
| OCDM | Vit-S | MAE | 75.42 | 45.18 | 56.03 | 41.53 | 41.26 | 29.68 |
| CUTER w/ $R_l$ | ResNet50 | Supervised | 74.35 | 46.80 | 46.34 | 37.15 | 38.30 | 22.53 |
| OCDM | ResNet50 | Supervised | 72.32 | 36.71 | 41.89 | 30.72 | 27.94 | 17.80 |
| CUTER w/ $R_l$ | ResNet50 | MoCo v3 | 75.04 | 48.31 | 47.20 | 37.48 | 40.06 | 23.41 |
| OCDM | ResNet50 | MoCo v3 | 75.31 | 47.05 | 42.45 | 30.30 | 33.45 | 21.08 |
| CUTER w/ $R_l$ | ResNet50 | DINO v1 | 76.43 | 50.42 | 48.52 | 40.35 | 40.42 | 23.79 |
| OCDM | ResNet50 | DINO v1 | 75.09 | 48.42 | 46.31 | 33.17 | 39.28 | 21.55 |
| CUTER w/ $R_l$ | ResNet50 | MAE | 73.62 | 45.14 | 43.52 | 36.68 | 37.45 | 22.15 |
| OCDM | ResNet50 | MAE | 74.25 | 44.97 | 42.13 | 35.90 | 38.07 | 21.94 |

the graph adjacency matrix requires maintaining and calculating gradients for multiple foreground object predictions. These factors create a trade-off between model throughput and performance. Future work will focus on developing acceleration techniques and adaptive processing strategies based on data stream velocities.

Finally, we visualize the localization capabilities of our CUTER model during multi-object continual learning on PASCAL VOC. We evaluate this from two perspectives: first, by calculating the average Fiedler value of the constructed feature graphs; second, by measuring the zero-shot localization performance ($AP_{50}$) between the generated bounding boxes and their corresponding ground truth for each incremental task. As shown in Figure 8, as training progresses, the connectivity of constructed graphs from streaming data gradually decreases while their separability improves. This enhancement in graph separability correlates with increased accuracy of the generated pseudo bounding boxes.

## F. Limitation

In the main text, our discussions primarily focus on scenarios using the ViT backbone as we need the structure like image patch to form feature patch which help us construct the graph. While CNN-based backbones like ResNet can also construct similar image patches, as shown in Appendix E.2, CUTER's performance somewhat degrades in these cases. Additionally, we acknowledge that performing cut-out operations introduces additional computational overhead as shown in Figure 8, which may sometimes affect the model's generalizability in online settings. However, as a pioneering work investigating

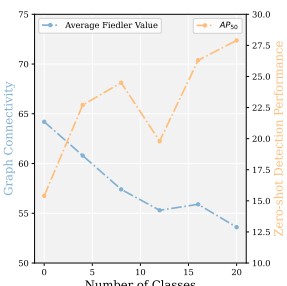

*Figure 8.* **Left:** Average model throughput comparison across 9 methods evaluated on 6 datasets. **Right** Evolution of localization performance: tracking the localization capabilities of our proposed CUTER model on PASCAL VOC from two perspectives.

object localization in MOCL, these limitations also point to promising directions for our future research.

