# OpenReview forum: "Cut out and Replay: A Simple yet Versatile Strategy for Multi-Label Online Continual Learning"
_ICML.cc/2025/Conference — ICML 2025 poster_

### Official Review · Reviewer_Nhoa · 2025-02-24

**Overall Recommendation:** 4

**Summary:**

This paper tackles Multi-Label Online Continual Learning (MOCL) from a novel perspective. Unlike existing approaches that focus on challenges like class imbalance and missing labels, this paper first analyzes the localization capabilities inherent in pre-trained models and introduces a CUT-out-and-Experience-Replay (CUTER) strategy. CUTER works by extracting, segmenting, and replaying image regions corresponding to individual labels. Through this approach, it addresses three key challenges simultaneously: catastrophic forgetting, missing labels, and class imbalance. The method's effectiveness is validated through comprehensive experimental results.

## update after rebuttal
After reading the authors' repsonses, I decide to keep my original score.

**Claims And Evidence:**

**Three important claims made in this paper:**
1. Modern pretrained models possess the innate localization capacity among which the multi-crop contrastive learning methods demonstrate the best localization potential.

2. The proposed regularization term on the feature-constructed graph serves to consolidate and further enhance model's localization capacity

3. The proposed CUTER strategy can simultaneously addresses catastrophic forgetting, missing labels, and class imbalance.

**Evidence**
For Claim 1, the paper provides a comprehensive analysis with clear visualization in Section 2.1, complemented by detailed quantitative comparison results across different pre-trained models in Table 7 (Appendix E.2). This systematic evaluation offers valuable insights into model selection for the proposed approach.

For Claim 2, theoretical derivation and analysis are presented in Section 2.3, supported by mathematical proofs and derivations in Appendix D.2. Conclusively, this paper establishes a strong theoretical connection between localization ability and the spectral norm of the feature-constructed graph. Certainly, it's also worth noting that the mechanism by which the proposed regularization improves model performance relies on additional assumptions.

For Claim 3, the comparison in Figure 4, Figure 5 and experimental results in Section 3 can demonstrate the method's effectiveness. The ablation studies in Section 3.3 are also informative.

**Essential References Not Discussed:**

I cannot think of essential references that are missing from this paper.

**Experimental Designs Or Analyses:**

**Experimental Designs**
As I stated in **Evaluations** , the experimental settings and evaluation protocols align well with established practices in related prior works. I did not find any inappropriate experimental design choices.

**Analyses**
The analyses for the three claims mentioned above are well structured with either empirical or theoretical verifications. A suggestion is that the ablation study could be further enriched by including specific runtime comparisons to quantitatively showcase the computational benefits of the proposed re-balanced sampling strategy, considering that Figure 8 only presents the combined running time of all components, which cannot reflect the claimed reduced computational cost.

**Methods And Evaluation Criteria:**

**Method**
This paper introduces CUT-out-and-Experience-Replay (CUTER), which uniquely addresses MOCL challenges by extracting, segmenting, and replaying label-specific image regions. By focusing on foreground object extraction, CUTER presents a novel direction that distinctly differs from conventional MOCL approaches [1,2,3], demonstrating notable innovation within this research domain.

The incorporation of nuclear norm regularization on the feature-constructed graph is noteworthy, providing both theoretical guarantees and a reasonable solution to the problem.

**Evaluations**
The method is evaluated on three established multi-label benchmarks using standard metrics (mAP, CF1, OF1). The comprehensive experiments, including ablation studies and comparative analyses with recent methods, convincingly validate CUTER's effectiveness both as a standalone approach and as a plug-in component. To further strengthen the evaluation, the authors could explore additional incremental settings with larger base class sets, similar to prior MOCL works [1,2,3].

[1] Knowledge restore and transfer for multi-label class-incremental learning. ICCV2023.
[2] Confidence self-calibration for multi-label class-incremental learning. ECCV2025.
[3] Optimizing class distribution in memory for multi-label online continual learning.

**Other Comments Or Suggestions:**

Some typos exist like "superiority of proposed our method" in line 39.

**Other Strengths And Weaknesses:**

**Strengths**
1. Multi-Label Online Continual Learning is an important and compelling research area. Furthermore, object region identification represents a fundamental challenge within this domain.
2. The analysis of localization capabilities across pre-trained models merits deeper exploration. The evaluation of these capabilities could be expanded beyond NCut and MCut metrics, as these represent just one category among many unsupervised and weakly-supervised object detection and segmentation approaches. A broader assessment incorporating diverse evaluation metrics would provide more comprehensive insights.
3. This paper presents an intuitive and straightforward approach that distinctly differentiates itself from existing work focused on class imbalance or missing labels. While taking this different direction, CUTER appears to address the fundamental challenges of MOCL more directly compared with strategies like pseudo labeling or rebalanced sampling, offering a more essential solution to the core problems in this domain.
4. The paper's use of spectral theorem analysis to examine localization capabilities across pre-trained models offers valuable insights. This theoretical foundation naturally leads to the intuitive development of low-rank regularization on the feature adjacency matrix, creating a cohesive analytical framework.

**Weaknesses**
1. Some related work exists in continual object detection shares quite similar intuition with this submission. Including comparisons and analysis with these works would better highlight this paper's unique challenges and innovations.
2. The analysis of different regularization terms warrants inclusion in the main text, as it constitutes a significant component of the paper's contribution.

**Questions For Authors:**

See the previous parts.

**Relation To Broader Scientific Literature:**

The paper's main contributions lie in advancing multi-label learning and its integration with online continual learning.

**Theoretical Claims:**

The paper's key theoretical analysis examines the relationship between the Fiedler value and noise matrix norm, grounded in assumptions about ideal feature-constructed graphs.

I have checked most of their proofs. Though not overly complex, they support the claims made in the main text and are clearly demonstrated.

---

> ### Author Rebuttal · Authors · 2025-03-30
>
> _Respected Reviewer Nhoa,_ we first thank you for your valuable and insightful feedback, and for recognizing the analysis and advantages of our proposed method. Below, we address your concerns in a point-by-point manner.
>
> Q: **Experiments on other incremental settings should be included.**
> A: We appreciate this suggestion. We have conducted additional experiments on different incremental settings following protocols established in previous works like KRT and OCDM. Specifically, we evaluated our approach on class-incremental scenarios with varying numbers of classes per stage:
>
> _MSCOCO B40-C10:_
> In this setting, we first train a base model on 40 classes, then incrementally add the remaining 40 classes over 4 sessions (10 classes per session).
>
> |   COCO (mAP)   | 1-40 | 1-50 | 1-60 | 1-70 | 1-80 |
> | :----:| :----: | :----: | :----: | :----: | :----: |
> | ER   | 69.8 | 54.2 | 50.7 | 44.8 | 36.4 |
> | PRS  | 69.3 | 56.5 | 52.0 | 44.7 | 39.8 |
> | OCDM | 65.8 | 52.1 | 50.4 | 48.2 | 37.3 |
> | KRT  | 68.4 | 57.3 | 52.1 | 46.5 | 40.0 |
> | CUTER| 69.0 | 59.6 | 57.4 | 54.7 | 50.8 |
>
> _NUSWIDE B41-C5:_
> Here, we start with 41 base classes and incrementally learn 8 sessions with 5 new classes each.
>
> |   NUSWIDE (mAP) | 1-41 | 1-46 | 1-51 | 1-56 | 1-61 | 1-66 | 1-71 | 71-76 | 1-81 |
> | :----:| :----: | :----: | :----: | :----: | :----: |:----: |:----: |:----: |:----: |
> | ER   | 64.4 | 51.2 | 47.3 | 40.0 | 35.8 | 31.3 | 28.8 | 29.5 | 24.6 |
> | PRS  | 61.2 | 50.6 |44.9 | 37.0 |36.7  | 30.2 |31.4 | 32.0 | 26.3 |
> | OCDM | 64.5 | 50.8 | 43.7 | 36.5 | 33.4 | 32.9 |31.8 | 30.8 |30.4 |
> | KRT  | 62.3 | 52.8 | 48.5 | 34.6 | 35.8 | 33.4 | 31.9 | 31.0 | 29.4 |
> | CUTER| 63.0 | 54.3 | 50.0 | 42.3 | 40.8 | 38.1 | 36.8 | 35.4 | 32.8 |
>
> These results demonstrate that our CUTER method consistently outperforms existing approaches across different incremental settings, particularly in later stages where catastrophic forgetting is most pronounced.
>
> Q: **Method in continual object detection should be compared and discussed.**
> A:Thank you for your insightful comment regarding the similarities between our proposed cut-out and replay framework and continual object detection. We have conducted a preliminary literature search on this topic and will include a thorough comparison in the related works section, especially with reference [A], which adopts an approach similar to our CUTER framework.
>
> However, we would like to clarify that methods from continual object detection cannot be directly applied to MOCL (Multi-Object Continual Learning). This is because these methods fundamentally rely on labeled bounding boxes which, even if partially missing, enable the training of an additional detection head. Such a strategy is clearly challenging to implement in the MOCL setting where such annotations are unavailable.
>
> To validate this point, we conducted a comparative experiment on the VOC dataset between our method and a DETR detection head using the box replay approach from [A] (the initial box was obtained by NCut). The results, as shown in the table below, demonstrate the advantages of our approach in the MOCL context where bounding box annotations are not accessible.
>
> |       | Avg mAP | Avg CF1 | Avg OF1 |
> | :----:| :----: | :----: | :----: |
> | ABR   | 71.51 | 62.52 | 65.49 |
> | CUTER | 82.07 | 72.19 | 75.27 |
>
> [A] Augmented Box Replay: Overcoming Foreground Shift for Incremental Object Detection. ICCV 2023.
>
> Q: **The analysis of different regularization terms warrants inclusion in the main text.**
> A: We agree this analysis provides important insights into our method. In the revised version, we'll include a concise comparison of regularization terms in the main text, explaining why nuclear norm regularization outperforms sparse and smooth approaches for preserving localization capabilities in MOCL. This addition will strengthen the theoretical justification for our design choices while maintaining the paper's focus.
>
> Q: **Ablation study could be enriched by including runtime comparisons between other methods and the proposed re-balanced sampling strategy.**
> A: We have enriched our ablation study by including runtime comparisons between our proposed re-balanced sampling strategy and other methods. For a fair comparison, we directly omit the CUT out process and measure the throughput in terms of samples processed per second:
>
> |   Implemented on RTX4090    | ER (random sampling) | PRS | OCDM | Ours |
> | :----:| :----: | :----: | :----: | :----: |
> | # samples per second   | 138.6 | 76.3 | 51.6 | 90.9 |
>
> Our proposed method delivers a strong throughput of 90.9 samples per second, which is considerably more efficient than both OCDM and PRS.
>
> Q: **Typos.**
> A: We feel sorry for any confusion or inconvenience may have caused. In the revised version, we will carefully proof-read our text to ensure that no grammar mistakes or typos still exist.

---

### Official Review · Reviewer_NiGx · 2025-03-06

**Overall Recommendation:** 3

**Summary:**

The paper tackles Multi-Label Online Continual Learning (MOCL) problem through a novel two-step approach. First, it identifies object-specific regions corresponding to labeled samples within each learning phase. Then, it selectively replays these regions, effectively circumventing the challenging missing label issue. The method also mitigates potential class imbalance problems in MOCL by transforming multi-label classification into single-label tasks. Experimental results across multiple image benchmarks validate its effectiveness.

**Claims And Evidence:**

**Important claims made in this paper:**
1. Multi-crop consistency pre-training enhances innate localization capabilities, whereas reconstruction-based training tends to promote feature sharing during recovery, which may hinder the effectiveness of spectral clustering-based localization approaches.
2. The Fiedler value of the feature graph has a direct upper bound determined by the norm of the perturbation matrix.

For claim 1, authors substantiate their assertions by examining the correlation between the average Fiedler values of features on the VOC dataset and zero-shot detection performance across different pre-trained models. Additional analysis in Appendix D.1 further corroborates these findings.

For claim 2, the authors provide a rigorous theoretical analysis to substantiate their claim.

**Essential References Not Discussed:**

See the weaknesses

**Experimental Designs Or Analyses:**

The experimental methodology appears sound, as the authors follow standard evaluation practices on visual classification benchmarks, enabling fair comparisons with MOCL and MLCIL methods. The empirical validation is comprehensive, including well-structured ablation studies examining the memory buffer's data distribution and demonstrating how the regularization effectively preserves the pre-trained model's inherent localization capabilities.

One limitation is that compared to other MOCL papers that evaluate multiple continual learning scenarios, this work primarily focuses on cases with similar number of classes across incremental stages. Including additional experiments with varying numbers of classes per stage would provide a more comprehensive evaluation of the method's robustness.

**Methods And Evaluation Criteria:**

This paper addresses the Multi-Label Online Continual Learning (MOCL) problem through a novel two-step approach. First, it identifies object-specific regions corresponding to labeled samples within each learning phase. Then, it selectively replays these regions, effectively circumventing the challenging missing label issue.

The proposed method is evaluated using standard protocols on multiple visual classification datasets, achieving state-of-the-art performance compared to existing MOCL and MLCIL methods. The authors further validate the effectiveness of each component and demonstrate its plug-and-play capabilities through additional experimental studies.

**Other Comments Or Suggestions:**

NA

**Other Strengths And Weaknesses:**

Strengths:
1. The topic of learning from online continual data streams is significant and represents a more realistic extension of traditional continual learning scenarios.
2. The proposed method demonstrates strong empirical performance, with comprehensive ablation studies effectively illustrating the function of each component.
3. The proposed regularization term is intuitive, and analyzing MOCL model degradation during continual learning from a spectral graph theory perspective offers an interesting theoretical insight.

Weaknesses:
1. Additional spectral clustering literature citations would help better contextualize how the regularization term functions. Moreover, since this work employs a detect-then-learn approach, whether other efficient detection methods could be integrated into the proposed MOCL framework.
2. Additional explanation is needed to clarify why the proposed regularization terms outperform conventional approaches like smoothing or sparse regularizations.

**Questions For Authors:**

What about the model's performance in an offline setting?

**Relation To Broader Scientific Literature:**

There is nothing in particular worth mentioning.

**Theoretical Claims:**

The authors establish a theoretical relationship between a graph's Fiedler value and the norm of the noise component in its graph Laplacian. Upon examination of the proof of Theorem 2.3 in the Appendix, the mathematical derivation and reasoning appear to be rigorous.

---

> ### Author Rebuttal · Authors · 2025-03-30
>
> _Respected Reviewer NiGx,_ we thank you for your valuable and insightful feedback. Below, we address your concerns in a point-by-point manner.
>
> Q: **Additional experiments with varying numbers of classes per stage should be included.**
> A: We appreciate this suggestion. We have followed the setting in previous works like KRT and OCDM to provide class-incremental settings with varying numbers of classes per stage. Due to the limited space, please refer to our response to Reviewer Nhoa's Q1 for complete tables.
>
> _MSCOCO B40-C10:_
> In this setting, we first train a base model on 40 classes, then incrementally add the remaining 40 classes over 4 sessions (10 classes per session).
>
> |   COCO (mAP)   | 1-40 | 1-50 | 1-60 | 1-70 | 1-80 |
> | :----:| :----: | :----: | :----: | :----: | :----: |
> | PRS  | 69.3 | 56.5 | 52.0 | 44.7 | 39.8 |
> | KRT  | 68.4 | 57.3 | 52.1 | 46.5 | 40.0 |
> | CUTER| 69.0 | 59.6 | 57.4 | 54.7 | 50.8 |
>
> _NUSWIDE B41-C5:_
> Here, we start with 41 base classes and incrementally learn 8 sessions with 5 new classes each.
>
> |   NUSWIDE (mAP) | 1-41 | 1-46 | 1-51 | 1-56 | 1-61 | 1-66 | 1-71 | 71-76 | 1-81 |
> | :----:| :----: | :----: | :----: | :----: | :----: |:----: |:----: |:----: |:----: |
> | PRS  | 61.2 | 50.6 |44.9 | 37.0 |36.7  | 30.2 |31.4 | 32.0 | 26.3 |
> | KRT  | 62.3 | 52.8 | 48.5 | 34.6 | 35.8 | 33.4 | 31.9 | 31.0 | 29.4 |
> | CUTER| 63.0 | 54.3 | 50.0 | 42.3 | 40.8 | 38.1 | 36.8 | 35.4 | 32.8 |
>
> These results demonstrate that our CUTER method consistently outperforms existing approaches across different incremental settings, particularly in later stages where catastrophic forgetting is most pronounced.
>
> Q: **Additional spectral clustering literature should be discussed and cited.**
> A: We appreciate this suggestion. In the revised version, we will expand our related work to include seminal spectral clustering works such as Shi and Malik (2000) on normalized cuts, Von Luxburg's (2007) tutorial on spectral clustering fundamentals, and more recent advancements like Tang et al. (2018) on robust spectral clustering. We'll also discuss image segmentation applications of spectral clustering beyond MCut, including works like LSC by Li and Chen (CVPR 2015) and the analysis of limitations of these spectral-based segmentation methods by Boaz Nadler (NIPS 2006).
>
> Q: **Whether other detection methods could be integrated into the MOCL**
> A: Yes, our CUTER framework is designed with modularity in mind, allowing for integration of various detection methods beyond MCut. However, we want to emphasize that in the online continual learning scenario, MCut's advantage of requiring no training is significant. While other detection methods like LOST, TokenCut, or attention-based approaches could theoretically be integrated, they would need to overcome these online learning constraints.
>
> Additionally, methods like SAM could be a viable choice, but without additional prompts, SAM typically produces many more segmentation masks than required for MOCL objectives. Its segmentations tend to be relatively fragmented because SAM's results are not inherently class-oriented or semantically guided. This would introduce additional challenges in establishing the correct correspondence between regions and labels.
>
> We greatly appreciate the reviewer's suggestions and will carefully discuss these considerations in a future version of our work.
>
> Q: **Additional explanation is needed to clarify why the proposed regularization terms work and outperform conventional approaches like smoothing or sparse regularizations.**
> A: To clarify why nuclear norm regularization outperforms conventional approaches: While all regularization methods in Table 3 are established in graph structure learning, they differ fundamentally in their effects on representation learning for our task.
>
> Nuclear norm regularization promotes low-rank structure in the graph Laplacian, better preserving the intrinsic manifold structure of ViT features while removing noise $\epsilon$ as established in Theorem 2.3. In contrast, sparse regularization, though effective for denoising, penalizes hypernode patches with naturally high cross-node similarity, disrupting inherent structural properties of ViT parameters and compromising classification capacity (Table 3). Similarly, smooth regularization forces excessive feature similarity across nodes, impairing the spectral clustering processes essential for effective localization.
>
> Q: **The model's performance in offline setting should be included.**
> A:Despite the offline setting not being the primary goal of our proposed method, we make a simple comparison with KRT (method designed for offline setting) on MSCOCO:
>
> |       | Avg mAP | Last mAP | Last CF1 | Last OF1 |
> | :----:| :----: | :----: | :----: | :----: |
> | KRT   | 75.7 | 69.3 | 63.9 | 64.7 |
> | CUTER | 84.1 | 76.5 | 70.3 | 73.4 |
>
> These results suggest that CUTER's design principles are broadly applicable across different multi-label classification settings.

---

### Official Review · Reviewer_eZ9b · 2025-03-09

**Overall Recommendation:** 4

**Summary:**

In this work, authors concentrate on Multi-Label Online Continual Learning (MOCL), mainly focusing on three main challenges: catastrophic forgetting, missing labels, and imbalanced class distributions. They introduce CUTER (CUT-out-and-Experience-Replay), a method that identifies and utilizes label-specific regions in images. By first evaluating pre-trained models' localization abilities and then implementing a region-based experience replay strategy, CUTER provides targeted supervision signals. Experimental results across multiple image benchmarks validate its effectiveness.

**Claims And Evidence:**

**Claims**
1. CUTER effectively addresses multiple MOCL challenges - catastrophic forgetting, missing labels, and class imbalance - while improving model performance.
2. The method achieves competitive results and can be readily integrated with existing approaches as a complementary component.
3. The model's performance benefits from the proposed localization-based regularization strategy.

**Evidence**
Regarding the paper's claims:

While CUTER's approach to addressing multiple MOCL challenges through region localization before learning and replay follows logically from its design, the paper would benefit from a more thorough discussion of how the accuracy and reliability of this localization process is ensured.
The paper effectively supports its second and third claims through both visual evidence (Figure 5) and quantitative results (Tables 3 and 5), demonstrating CUTER's performance advantages and successful integration with existing methods.

**Essential References Not Discussed:**

N/A

**Experimental Designs Or Analyses:**

The evaluation methodology aligns with established practices in Multi-label learning and MOCL, utilizing multi-label benchmarks and standard metrics (mAP, CF1, OF1).

Additionally, the authors present an interesting connection between pre-trained models' MOCL capabilities and their derived feature characteristics. The experimental results, particularly the analysis of different regularization methods and subsequent investigations, provide empirical support for their findings.

**Methods And Evaluation Criteria:**

This paper addresses MOCL through label-specific feature learning, introducing a novel label-attentional mechanism to identify and replay label-specific image regions. Leveraging state-of-the-art pre-trained models, the approach incorporates a regularization technique based on spectral graph theory principles commonly used in unsupervised segmentation methods. The method represents an innovative departure from existing MOCL approaches (which typically focus on memory buffer sampling or pseudo labeling techniques) and demonstrates both novelty and comprehensive design.

The evaluation methodology aligns with established practices, utilizing multi-label benchmarks and standard metrics (mAP, CF1, OF1). The authors provide thorough empirical validation through ablation studies demonstrating the effectiveness of individual components, as well as results showing successful integration with existing methods. The experimental results effectively support the paper's claims.

**Other Comments Or Suggestions:**

See the previous parts.

**Other Strengths And Weaknesses:**

Strengths
1. The proposed method offers a novel approach to the problem. The authors present a clear methodology that directly addresses label-region correspondence - a challenging yet fundamental issue in the field. This approach stands out from existing MOCL methods and recent works on multi-label classification with missing labels.
2. The proposed regularization term is well-motivated and theoretically validated. The analysis raises an interesting question about whether similar metrics, such as the averaged Fiedler value, could be applied to other domains like pre-training task design or dataset evaluation.
3. The method demonstrates strong empirical performance across multiple datasets. Through comprehensive ablation studies and informative visualizations, the authors effectively illustrate the contribution of each component to the overall system.

Weaknesses
1. As stated by the authors, performing cut-out operations and further regularization terms to consolidate model's localization capacity introduces additional computational overhead.
2. There are some typos exist in the main text.
3. In Section 2.3, while the authors introduce nuclear norm regularization for the derived graph Laplacians, the underlying mechanism is relegated to the appendix, and detailed comparisons are deferred to the experimental section. A more cohesive presentation of this material within the main text would enhance the section's clarity and impact.

**Questions For Authors:**

N/A

**Relation To Broader Scientific Literature:**

As authors stated in the concluding paragraph, this research contributes to advancing machine learning methodology. While acknowledging that the work may have broader societal implications, an in-depth discussion of specific impacts falls outside this paper's technical scope.

**Theoretical Claims:**

The paper's key theoretical contribution centers on analyzing the model's localization capability through its relationship to the adjacency matrix constructed from learned features. After reviewing the proof of Theorem 2.3, I find the mathematical reasoning to be logically sound.

---

> ### Author Rebuttal · Authors · 2025-03-30
>
> _Respected Reviewer eZ9b,_ we first thank you for your valuable and insightful feedback, and for recognizing our motivation and theoretical analysis. Below, we address your concerns in a point-by-point manner.
>
> Q: **How the the accuracy and reliability of proposed localization process is ensured?**
> A: Our approach ensures the accuracy and reliability of the proposed localization process through three key mechanisms:
>
> 1. **Principled Model Selection**: We conducted detailed analysis of the localization potentials across different pre-trained models, establishing general selection principles that identify models with optimal localization capabilities for MOCL tasks.
>
> 2. **Selective Label-Region Matching Strategy**: We developed a targeted matching approach that accurately aligns semantic regions with corresponding labels, minimizing false correlations and enhancing localization precision.
>
> 3. **Spectral-based Regularization**: To prevent degradation of localization capacity during MOCL progression, we implemented a specialized regularization term that preserves the model's ability to accurately identify regions of interest over time.
>
> The effectiveness of these mechanisms is substantiated by both qualitative evidence (visualizations in Figures 3 and 5 demonstrating accurate region identification) and quantitative validation (ablation studies in Table 6 showing the benefits of our Cut-out Replay strategy in capturing fine-grained spatial information). Together, these elements form a comprehensive framework that consistently delivers accurate and reliable localization performance throughout the MOCL process.
>
> Q: **Performing cut-out operations and regularization terms introduce additional computational overhead.**
> A: We acknowledge that our approach introduces some additional training time. In fact, as shown in Figure 8 (left) in the Appendix, when not using regularization terms, CUTER achieves relatively comparable model throughput to other popular MOCL methods. Additionally, we recognize that the proposed regularization terms do cause a considerable decrease in model throughput. Finding ways to accelerate this process and achieve a better balance between performance and computational efficiency will be a focus of our future work.
>
> Q: **Nuclear norm regularization details spread across appendix and main text, affecting clarity.**
> A: We appreciate the reviewer's valid point about the nuclear norm regularization presentation. The current separation was primarily due to page constraints. In the revised version, we will restructure Section 2.3 to present a more cohesive narrative by:
> (1) Introducing the theoretical motivation behind nuclear norm regularization;
> (2) Connecting it directly to the derived graph Laplacians with concise mathematical formulations;
> (3) Providing intuitive explanations of how this regularization enhances representation learning;
> (4) Briefly highlighting comparative advantages over alternative approaches like smooth regularization or sparse regularization on the graph laplacian before the experimental section;
>
> Q: **Typos.**
> A: We feel sorry for any confusion or inconvenience may have caused. In the revised version, we will carefully proof-read our text to ensure that no grammar mistakes or typos still exist.

---

### Decision · Program_Chairs · 2025-05-01

**Decision:**

Accept (poster)

**Comment:**

The paper proposes CUTER (CUT-out-and-Experience-Replay), a novel strategy for Multi-Label Online Continual Learning (MOCL) that addresses key challenges including catastrophic forgetting, missing labels, and class imbalance. The method leverages pre-trained models' localization capabilities to identify label-specific regions in images, which are then selectively replayed to provide fine-grained supervision. CUTER incorporates a spectral graph theory-based regularization to preserve localization ability during continual learning, achieving state-of-the-art performance across multiple benchmarks while serving as an orthogonal solution that can integrate with existing approaches.

The reviewers unanimously recognized the paper's strengths: (1) Novelty and Technical Soundness. (2) Empirical Validation: All reviewers acknowledged the thorough experiments across three benchmarks. (3) Practical Impact: Reviewers pointed out that CUTER addresses "fundamental challenges more directly" than prior methods like pseudo-labeling. While reviewers raised minor concerns about computational overhead (eZ9b) and presentation of regularization analysis (NiGx), the authors convincingly addressed these in rebuttals by providing additional runtime comparisons. All reviewers ultimately recommended acceptance. Given this consensus, I recommend acceptance.